# Interplay between Lignans and Gut Microbiota: Nutritional, Functional and Methodological Aspects

**DOI:** 10.3390/molecules28010343

**Published:** 2023-01-01

**Authors:** Simone Baldi, Marta Tristán Asensi, Marco Pallecchi, Francesco Sofi, Gianluca Bartolucci, Amedeo Amedei

**Affiliations:** 1Department of Experimental and Clinical Medicine, University of Florence, 50134 Florence, Italy; 2Department of Neurosciences, Psychology, Drug Research and Child Health, University of Florence, 50139 Florence, Italy; 3Unit of Clinical Nutrition, Careggi University Hospital, 50134 Florence, Italy; 4Interdisciplinary Internal Medicine Unit, Careggi University Hospital, 50134 Florence, Italy

**Keywords:** lignans, enterolignans, phytoestrogens, gut microbiota, GC-MS, HPLC

## Abstract

Lignans are non-flavonoid polyphenols present in a wide range of foods frequently consumed in the Western world, such as seeds, vegetables and fruits, and beverages such as coffee, tea and wine. In particular, the human gut microbiota (GM) can convert dietary lignans into biologically active compounds, especially enterolignans (i.e., enterolactone and enterodiol), which play anti-inflammatory and anti-oxidant roles, act as estrogen receptor activators and modulate gene expression and/or enzyme activity. Interestingly, recent evidence documenting those dietary interventions involving foods enriched in lignans have shown beneficial and protective effects on various human pathologies, including colorectal and breast cancer and cardiovascular diseases. However, considering that more factors (e.g., diet, food transit time and intestinal redox state) can modulate the lignans bioactivation by GM, there are usually remarkable inter-individual differences in urine, fecal and blood concentrations of enterolignans; hence, precise and validated analytical methods, especially gas/liquid chromatography coupled to mass spectrometry, are needed for their accurate quantification. Therefore, this review aims to summarize the beneficial roles of enterolignans, their interaction with GM and the new methodological approaches developed for their evaluation in different biological samples, since they could be considered future promising nutraceuticals for the prevention of human chronic disorders.

## 1. Introduction

Lignans, similarly to isoflavones, ellagitannins, stilbenes and coumestan, are plant-derived polyphenols that, having an estrogen-like chemical structure, act as phytoestrogens in the human body [1]. The main dietary source of lignans is represented by flaxseed, and their digestion by the gut microbiota (GM) produces more active and bioavailable metabolites named enterolignans [2]. These molecules exert (i) potent anti-oxidant and anti-carcinogenic activities and (ii) estrogen agonism/antagonism properties. In addition, recent evidence has highlighted their beneficial role in the prevention and treatment of menopausal symptoms, skin aging, osteoporosis, cancer and cardiovascular, neurodegenerative, immune and metabolic diseases [3]. However, some inter-individual variations could be observed in enterolignan production, mainly due to differences in diet habits, food transit time, intestinal redox state and GM composition and function, with consequent diverse enterolignans abundance in urine, fecal, and blood samples [4]. Thus, many efforts have been performed in the last years working towards the development of precise, robust, and validated analytical methods (mostly based on gas/liquid chromatography technology) for the qualitative and quantitative analysis of enterolignans in human biofluids. In this scenario, our review summarizes the health-promoting role of enterolignans, their interplay with GM and the methodological approaches that have been developed for their evaluation in different biological samples, because, in the future, lignan-rich foods could be considered promising nutraceuticals for the prevention of human chronic diseases and cancers. 

## 2. Food Sources and Nutritional Properties of Lignans

The term lignan, introduced in 1942 by Harworth, refers to a family of secondary metabolites, found in plants, which have been derived from the metabolism of phenylalanine [1]. Specifically, they are polyphenolic compounds resulting from the combination of two phenylpropanoid units, C 6–C 3, coupled at β and β′ carbon [2], and usually linked to other molecules as glycosylated derivatives [3]. Based on the incorporation of oxygen within cyclization patterns, lignans can be subdivided into eight structural subgroups: aryltetralins, arylnaphthalenes, dibenzocyclooctadienes, dibenzylbutanes, dibenzylbutyrolactones, dibenzylbutyrolactols, furans and furofurans [4,5]. Although new extraction approaches have made possible to isolate and identify a large number of lignans [6]; the lignans commonly distributed in food and for which there are more convincing evidence for their health benefits are: lariciresinol, matairesinol, pinoresinol, and secoisolariciresinol (SECO). In addition, other interesting lignans such as sesamin, syringaresinol, medioresinol, arctigenin or sesamolin can be highlighted [7,8]. 

In general, lignans are widely found in plant-based foods such as seeds, vegetables, cereals, legumes, nuts and fruits, and, to a lesser extent, beverages like tea and coffee. Flax and sesame seeds are among the foods with the highest lignan content, providing 284.00 mg/100 g and 776.49 mg/100 g, respectively. In particular, the flax seeds have a higher SECO and lariciresinol content, while sesame seeds have a higher sesamin and sesamolin content [9]. They are recognized as significant sources of lignans in the diet of different geographical areas, such as the Middle East, North Africa and Asia, but differences can be found between other regions depending on the dietary patterns of the population [10]. In fact, in Western countries, two of the main sources of lignans are cereals and grain products [11]. In Europe, the EPIC (European Prospective Investigation into Cancer and Nutrition) cohort study reported a difference between Mediterranean countries, where the central source of lignans was olive oil, while in non-Mediterranean countries it was cereals [12]. Both nutritional products contain less than 5 mg/100 g of lignans; olive oil is the principal food source of pinoresinol, while cereals contain lariciresinol, syringaresinol, and, in smaller quantities, matairesinol and SECO [9]. The lignans in cereals are mainly found in the outer layers and the aleurone layer, so they are present in whole grain products [13]. In addition to the processing degree, their content also varies according to the type of cereal, with rye and wheat containing the most lignans [14].

## 3. Gastrointestinal Digestion and Fermentative Process of Lignans

The biological properties of dietary phytoestrogens, including lignans, are limited by their low bioavailability [15]. 

In general, the final biological activity of all phytoestrogens depends on the GM metabolism. For instance, equol (4′,7-isoflavandiol) has been described as an important bacterial metabolite of daidzein (an isoflavone present in soy) with higher beneficial health effects; on the other hand, daidzein can also be converted by GM into the less active O-desmethylangolensin [16].

Similarly, plant lignans are transformed into enterolignans, namely enterolactone (EL) and enterodiol (ED), by GM [17]. These compounds, also named mammalian lignans, are more bioavailable compared to their precursors [18,19] and are associated with beneficial effects on human health [20].

Dietary lignans are mostly transformed when they reach the colon, as available evidence suggests some marginal alterations during their passage through the stomach and small intestine [3,18], with a very low ratio between ingested plant lignans and aglycone absorption in the small intestine [21]. A significant proportion of lignans reaching the colon are metabolized following a multi-step process by various bacterial communities, including deglycosylation, demethylation, dehydroxylation and dehydrogenation [22]. Paradigmatic for this process is the commonly described SECO metabolism because it is the intermediate aglycone form of most plant lignans (Figure 1) [23].

The initial step involves deglycosylation, which consists of the hydrolysis of the sugar from secoisolariciresinol diglucoside (SDG) to obtain the aglycone SECO. This reaction is carried out in anaerobic conditions by members of *Bacteroidetes* (*B. distasonis*, *B. fragilis* and *B. ovatus*) and *Clostridium* (*C. cocleatum* and *C. saccharogumia*) genera [24]. The second step is the O- demethylation of SECO to produce dihydroxy-enterodiol (DHEND) and is mainly carried out by *Butyribacterium methylotrophicum*, *Eubacterium* spp. (i.e., *E. callanderi* and *E. limosum*), *Peptostreptococcus productus* and *Blautia producta*. On the third step, the dehydroxylation of DHEND into ED is mediated by *Eggerthella lenta* and Clo*stridium scindens* [18,25,26]. Lastly, *Ruminococcus* spp. determines the dehydrogenation of ED into EL, the final product of the pathway [27,28]. Moreover, another alternative route for EL synthesis without the production of the intermediate ED consists in the production of dihydroxy-enterolactone (DHENL) through the dehydrogenation or oxidation of DHEND [29] or via the demethylation of mataresinol [15,30]. These reactions are respectively catalyzed by *Lactonifactor longoviformis* and *Ruminococcus productus* and the subsequent DHENL dehydroxylation by *Eggerthella* spp. results in EL production [24,26,31].

Once ED and EL are produced, some studies suggest that they are absorbed by colonic epithelial cells and conjugated to glucuronide acid or sulfates, before entering enterohepatic circulation [32,33]. Several findings have demonstrated a positive correlation between lignans intake and plasma levels of enterolignans [34] but, although EL is more rapidly absorbed than ED, no blood enterolignans values are detected until 8–10 h after dietary intake [35], and the peak serum concentrations of ED and EL are found to be 12–24 h and 24–36 h, respectively [36]. Finally, the enterolignans are mainly excreted as monoglucuronides in urine but small amounts of diglucuronides and sulfates also occur; conversely, free enterolignans are excreted through feces. Nevertheless, although a plant-based diet centered on whole grains, vegetables and fiber results in higher levels of circulating and excreted enterolignans [17], but some inter-individual variability in EL production has been observed. The main factors that may interfere with enterolignans production include genetics, sex, age, food transit time, intestinal redox state, smoking habit, antibiotic uptake and GM composition and function [6,35,37,38,39]. In particular, the two-way interaction between polyphenols and GM (modulation of the microbiota by polyphenols and metabolism of polyphenols by the microbiota) has been recently proposed as the main driver of the inter-individual variation, assuming the existence of different GM-associated metabotypes [40]. For instance, it is well documented that the microbial dehydrogenation of ED to generate EL is a critical step in plant lignans metabolism that could explain the greater variation in enterolignan production among individuals [35,37,41]. However, further research is needed to better understand the distribution and posterior metabolism of enterolignans in humans, as most of the available evidence comes from preclinical data from animal models [42].

## 4. Biological Activities and Health-Promoting Effects of Lignans

Lignans, enterolignans and, in general, phytoestrogens can bind to estrogen alpha and beta receptors thanks to their structural similarity with 17β-estradiol, enabling them to affect all the processes regulated by estrogens. For instance, they increase the levels of sex hormone binding globulin, which limits the diffusion of sex steroids into target tissues modulating their bioactivity; they activate the transcription of specific target genes and they promote cell proliferation/apoptosis in different tissues (e.g., reproductive, skeletal, cardiovascular and central nervous systems) [43,44]. Additionally, enterolignans play other biological effects not mediated by estrogen-receptor (ER), such as (i) the immune system modulation by acting on NF-kB signaling [43], (ii) the activation of serotoninergic and IGF-1 (insulin-like growth factor 1) receptors [45,46], (iii) the induction of DNA methylation and histone modification [47], (iv) the regulation of tyrosine kinase cascades [48], and (v) the inhibition of both lipids’ peroxidation and oxygen species’ production [49,50].

Furthermore, given the aforementioned structural similarities between polyphenols and estrogens, lignans also inhibit ER activity (anti-estrogenic action) and ER expression and consequently reduce cell proliferation [51].

Hence, these hormonal and non-hormonal properties of enterolignans, as supported by recent findings, are probably responsible for their important anti-oxidant, anti-proliferative, anti-inflammatory, anti-mutagenic and anti-angiogenic proprieties that have multiple beneficial effects on health promotion and disease prevention in humans (Figure 2).

For instance, concerning anti-inflammatory activity, some lignans have the capacity to inhibit NF-kB activity on human mast cells (HMC-1), thus reducing the production of pro-inflammatory cytokines [52]. Furthermore, lignans are able to suppress nitric oxide (NO) generation and decrease inflammatory cell infiltration, but, on the other hand, although a free radical formation is an inevitable by-product of cellular metabolism, the accumulation of reactive oxygen species can damage cellular constituents and play an important role in the pathogenesis of various severe disorders [53]. However, many studies have demonstrated the strong anti-oxidant activity of plant extracts, which is mainly attributable to the highly-effective anti-oxidant properties t of lignans (and polyphenols in general), thus confirming their potential use as preventive and/or therapeutic clinical tools [12].

The currently available evidence is either obtained through epidemiological studies assessing the intake of lignans using food frequency questionnaires and their subsequent analysis through specific databases, or by quantifying markers of consumption such as enterolignan levels in serum, blood or urine [44,45]. These data link the consumption of lignan-rich diets with a reduced developing risk on various pathologies, such as cardiovascular disease and different types of tumors, especially breast and colorectal cancers [46,47,48,49].

### 4.1. Cancers

The scientific interest in the anti-cancer effects of lignans and their precursors is growing, as pre-clinical research has suggested that EL could prevent cancer progression through the inhibition of inflammation, tumor growth, angiogenesis, metastasis and the induction of apoptosis in cancer cells [50]. Breast and colorectal (CRC) cancers are some of the tumors for which the protective lignans’ role has been most documented. Since some types of breast tumors are hormone-sensitive, a potential mechanism of lignans may be due to ER binding, but also via inhibition and downregulation of HER2 (human epidermal growth factor receptor 2) and EGFR (epidermal growth factor receptor) [54,55]. Clinical trials have reported that high EL levels are associated with a lower risk of breast cancer; although some studies have shown this association only in premenopausal women [56], others have shown the relation in postmenopausal women [57,58,59]. Higher lignans consumption was inversely correlated with the risk of ER-negative breast carcinoma among premenopausal women and with the risk of ER-positive breast cancer among post-menopausal women [60]. In accordance with these findings, a German case-control study has demonstrated a positive correlation between a high intake of lignan-rich seeds and the significant decrease of ER-positive breast cancer risk in post-menopausal women [61].

In general, a higher intake of lignans has also been linked with a better prognosis in patients with breast cancer because of their consistent anti-cancer effects determined by their ability to (i) reduce tumor volume and size by the induction of cancer cell apoptosis, (ii) reduce cancer cell derived vascular endothelial growth factor (VEGF) and inhibit estradiol (E2)-induced angiogenesis, (iii) decrease the in-vivo release of interleukin-1 beta, (iv) inhibiti the uPA/Plasmin/MMPs mediated extracellular matrix remodeling and (v) revert the TGF-β induced EMT (epithelial–mesenchymal transition) via the modulation of ERK-NFκB-Snail signaling pathway [46,50].

Similarly, the preventive effect of lignans in CRC could be related to ERβ, as its expression represents a possible mechanism linked to cancer progression [37,62]. However, further research is needed to well-define the effects of lignans on CRC, as the available results are more heterogeneous. Some studies have found no association between EL concentrations and CRC risk [48,63,64], while others have found that increased EL levels are associated with a reduced risk of CRC [65]. Notably, a couple of studies found a lower risk only in women, and in one of them the men were associated with an increased risk [66,67]. Furthermore, lignan consumption has also been associated with a reduced risk of gastro–esophageal and prostate tumors, but few human studies have been conducted. Regarding gastro–esophageal cancer, conflicting findings have been found; some research has in fact documented that lignans intake decreased the tumor risk [68,69] while another study, examining the Swedish Cancer Registry database, did not find a clear association between dietary lignan consumption and development of gastro–esophageal carcinoma [70]. Likewise, while some studies have documented an association between lignans intake and reduced prostate cancer risk, i.e. due to the EL capability to induce apoptosis in human prostate carcinoma LNCaP cells by inhibiting Akt signaling pathway [71,72,73], others reported no significant associations between the incidence of prostate cancer and plasma EL levels [74,75].

### 4.2. Cardiovascular Diseases

Many studies support that a diet rich in lignans and higher EL levels are associated with a lower risk of cardiovascular diseases [76,77]. Part of the cardioprotective effects associated with lignans might be due to the prevention of major risk factors for heart diseases, such as low-grade inflammation, type 2 diabetes (T2D) and metabolic syndrome [78,79], through the modulation of the vascular reactivity by acting on smooth muscle cells as well as on the endothelium, unbending precontracted vessels, inhibiting of calcium entry, enhancing expression of nitric oxide synthase, stimulating of prostacyclin production and preventing of myointimal hyperplasia [80].

Results from a study evaluating two cohorts of U.S. women documented a relation between lignan metabolites, particularly EL, with a lower risk of T2D [81], while the NHANES (National Health and Nutrition Examination Survey) cohort study found that greater levels of EL were inversely associated with metabolic syndrome [82]. Benefits have also been reported on individual components of the metabolic syndrome, with higher EL levels being correlated with more favorable cardiometabolic risk factors, including abdominal obesity, blood pressure and levels of triglycerides, high-density lipoprotein (HDL) cholesterol and fasting glucose [82,83,84]. In addition, the lignans may also reduce markers of inflammation, although the conclusion is pending due to the high heterogeneity of the available results. In addition, the NHANES study has documented a significant inverse association between enterolignans and markers of chronic inflammation, in particular the lower levels of C-reactive protein (CRP) [85]. On the other hand, two meta-analyses that evaluated the efficacy of flaxseed found no reduction in CRP, or only a CRP reduction in obese patients, but instead reported a lower level of TNF-α [86,87].

### 4.3. Other Pathological Morbidities

Exposition of women to lignans and other phytoestrogens in the pre- and post-menopausal period may prevent the menopausal symptoms induced by declined endogenous estrogen production, vasomotor symptoms and vaginal atrophy and may serve as a treatment option for patients who have contraindications to hormone therapy [88]. Contrariwise, there are no positive effects of dietary phytoestrogen on males, in contrast to females, while the reproductive functions have not been demonstrated yet [89]. Moreover, the estrogens’ deficiency following menopause is usually responsible for atrophic skin changes, but some reports have demonstrated that lignans exert anti-aging effects promoting skin vascularization and cell proliferation [90]. In addition, as phytoestrogens can inhibit osteoclast differentiation but promote bone formation, they play an important role in bone remodeling [91]. Furthermore, in recent years, several findings have confirmed that the GM can produce neurotransmitters and neuropeptides that are capable of crossing the mucosal intestinal layer and reaching the brain via the so-called gut-brain axis, where they can modulate some cognitive functions [92,93]. Microbial-derived enterolignans have shown the ability to prevent neuroinflammation and neurodegeneration by the modulation of the gut-brain axis [94,95]. In detail, in addition to its acetylcholinesterase and butyrylcholinesterase activities, EL can inhibit the carbonic anhydrase [96] and attenuate the degeneration of the striatal dopaminergic terminals in Parkinson’s disease rat model [97].

## 5. Current Approaches for Qualitative and Quantitative Analysis of Lignans

Considering the recent increased consumption of polyphenols, especially lignans, primarily due to their health-promoting properties, large research efforts have been devoted in the last few years to developing several analytical methods for their qualitative and quantitative analysis in different human biofluids (Figure 3).

In general, lignans are mostly available in conjugated form, such as glucuronides and sulphates, so samples must be enzymatically hydrolyzed using β-glucuronidase/sulfatase preparations before extraction and chromatographic analysis [87]. Extraction is another important step in the isolation and identification of lignans and, to date, liquid-liquid extraction (LLE) and solid-liquid extraction (SLE) with solvents such as methanol, acidified methanol or a methanol–water mix are the most used. This phase usually involves a stirring step (performed by vortex, shaker or ultrasonic bath) that determines the direct extraction of polyphenolic compounds in samples; however, this procedure could be influenced by several factors, e.g., contact time, pH and temperature [88]. Subsequently, since solvent extraction implies the co-extraction of other non-phenolic substances, an additional step for their removal is usually carried out by solid phase (SPE) or LLE, to avoid potential interferences. Afterwards, the currently predominant method for the detection of lignans, enetrolignans and in general phytoestrogens is the high-performance liquid chromatography (HPLC) [88,89]; however, accurate and validated protocols have also been developed using gas chromatography mass spectrometry (GC-MS) [89,90,91,92], and time-resolved fluoro-immunoassay (TR-FIA) [93] (Table 1).

### 5.1. High-Performance Liquid Chromatography (HPLC)

Nowadays, given its high reproducibility, high sample recovery and reduced sample pre-treatment, HPLC represents the predominant analytical method for the quantitation of lignans. Oftentimes UV or UV–DAD (diode array detection) detectors have been used for the lignans determination in animal or human samples [92]. For instance, a reversed-phase HPLC–UV method, using p-Hydroxybenzophenone as an internal standard, has been proposed for the evaluation of free and conjugated lignans, following their extraction with dichloromethane [93]. However, more sensitive and selective approaches than HPLC-UV have been developed for reliable quantification of lignans in complex matrixes, such as HPLC coupled to CEAD (coulometric electrode array) or MS (mass spectrometry) detection methods. In general, the HPLC–CEAD technique is limited to lignans with free phenolic hydroxyl groups, while HPLC–MS is limited to compounds with ionizable groups. In detail, HPLC–CEAD, which allows the resolution of co-eluting analytes based on the differences in their oxidation-reduction behavior, has been applied for developing and validating a method for the quantification of EL, ED and other lignans in plasma as well as in the uterine tissue of rats, and in human plasma and urine samples [35,89,90,91,92]. For instance, Gamache and Acworth coupled a reversed-phase column with eight coulometric electrodes and used a mobile phase consisting of sodium acetate buffer, methanol and acetonitrile for the determination of urinary free phytoestrogens, previously hydrolyzed with β-glucuronidase [94]. On the other hand, for the quantification of plasma lignans, Nurmi and Adlercreutz pre-treated the samples with glucuronidase/sulfatase, extracted them with diethyl ether and, before the analysis with an HPLC–CEAD method, redissolved the evaporated samples in methanol [95]. Instead, in the first study in which lignans were quantified in urine samples using HPLC-APCI (atmospheric pressure chemical ionization) coupled to multiple-reaction monitoring (MRM)-MS, Horn and colleagues extracted lignans from samples using Sep-Pak C18 cartridges and adding 4-methylumbelliferone glucuronide as an internal standard. Subsequently, after hydrolysis with glucuronidase/sulfatase, aglycones were recovered by solid-phase extraction and analyzed with HPLC-APCI-MS using a C8 reversed-phase column and a solvent gradient of 0–50% acetonitrile in aqueous ammonium acetate over 15 min [91]. In addition, the same methodological approach was used for ED and EL quantification in human serum and urine, with results comparable to those obtained by HPLC-CEAD [92].

In the 2000s, other enterolignans quantification methods in both rat and human plasma or serum samples based on HPLC-ESI (electrospray ionization) coupled to MS/MS-MRM have been developed [91,92,93]. For instance, Knust and collaborators proposed an HPLC-ESI-MS protocol for the measurement of free enterolignans as well as their monoglucuronide conjugates in human biofluids with minimal sample preparation [94]. Recently, a rapid, reproducible, and sensitive LC−MS/MS method has been developed to quantify the circulating glucuronidated, sulfated, and free EL [96], and also a validated HPLC-DAD-ESI-MS method has been proposed for the quantification of sixteen different phytoestrogens in food and human (serum and urine) samples [97].

### 5.2. Gas Chromatography-Mass ThenSpectrometry (GC-MS)

GC–MS is also considered an affordable analytical technique with excellent chromatographic resolution for lignan analysis, and many different protocols have been proposed since the early 1990s. In general, for GC–MS analysis of lignans, the final column temperature is usually comprised between 280 and 300 °C, the carrier gas is traditionally helium, internal standard substances are added before extraction phase and the more used ionization technique is electron impact (EI); however, in absence of reference compounds, care should be taken during the interpretation of fragmentation patterns [98]. The first isotope dilution GC-MS method for the determination of ED, EL, and other lignans in urine samples has been proposed by Adlercreutz et al. [99]. The urine samples were extracted on Sep-Pak cartridges, conjugated fractions were isolated by chromatography on the acetate form of DEAE-Sephadex and deuterated internal standards were added to the samples before hydrolysis. Then, the hydrolysate was extracted on a Sep-Pak cartridge and, after chromatography on the acetate form of QAE-Sephadex and silylation, the samples were analyzed by GC–MS in the selective ion monitoring (SIM) mode. The same research group also presented the first method for the quantitative analysis of lignans in plasma samples [100]. The samples were separated in ion-exchange chromatography in two fractions, one constituted by free compounds and mono- and di-sulfates and the other containing the mono- and di-glucuronides and sulfo-glucuronides. Following the hydrolysis, the fractions were purified by solid phase extraction and ion exchange chromatography and, after adding deuterated internal standards, analyzed by GC/MS/SIM. Instead, Morton et al. proposed an isotope dilution method for lignans assessment in foods and human samples [101] consisting of their hydrolyzation and following separation by Sephadex LH-20; after, samples were derivatized using N,O-Bis(trimethylsilyl)trifluoroacetamide (BSTFA) and analyzed by GC/MS/SIM.

Lately, the development of GC–MS methods in micro-selected ion storage (μSIS) mode has improved ion sensitivity and selection by reducing interfering background noise; in addition, the incorporation of isotopically labelled compounds as internal standards has allowed an accurate quantitation thanks to the removal of any errors that may have occurred during sample preparation and GC–MS analysis. Therefore, Edel et al. have recently developed a SLE method for the isolation of plasma enterolignans, using 2H6-labeled isotopes, GC–MS in μSIS mode [102].

### 5.3. Immunoassays

Immunoassays are based on the principle of the antigen-antibody interaction; labelled and unlabelled antigens compete for binding to the limited number of antibody binding sites and the concentration of unlabelled antigen is derived from the extent to which it competitively inhibits the linking of the labelled antigen to a specific antibody. So, immunoassays usually provide an increase in the speed and sensitivity compared to GC–MS or HPLC but sometimes a decrease in selectivity results, since cross-reactivity can occur. Two decades ago, a time-resolved fluoro-immunoassay (TR-FIA) was developed for the rapid analysis of lignans, especially EL, in human urine and plasma samples [98,99,100]. Later, this method has been used for large population studies investigating the relationship between human serum or plasma EL concentration and the risk of specific disease [101,102]. Nevertheless, a study measuring the concentrations of EL enantiomers in the serum of adult volunteers consuming their habitual diet and after flaxseed ingestion reported that, in comparison with chromatographic quantitation methods (i.e., HPLC-MS/MS, HPLC-CEAD, GC-MS), TR-FIA underestimates EL concentrations [100].

Therefore, in general, ethers of lignans can be routinary separated and quantified by GC–MS even in the case of complex mixtures of lignans and other polyphenols; conversely, HPLC on reversed-phase columns is especially suited for analysis of lignans and their metabolites in biological matrixes. However, the recent development of HPLC-ESI-MS and corresponding techniques with high sensitivity and selectivity has proven valuable in lignan analysis, also allowing the separation of lignan enantiomers on chiral HPLC columns.

## 6. Conclusions and Future Perspectives

Although additional studies are needed to better understand the specific (cellular and molecular) mechanism of action of enterolignans, their beneficial health effects for various pathological conditions have been widely documented. Surely, taking into account that large sources of dietary lignans are available in all types of nutritional regimens around the world, a better understanding of the interplay existing between the GM and its metabolites (i.e., enterolignans) may pave the way for the use of lignans as nutraceuticals, in order to exploit their remarkable effects in preventing local and systemic disorders in humans. Of course, considering the well-established paramount importance of GM in the health outcomes of dietary polyphenols, a better characterization of the specific bacterial strains that can metabolize lignans in their effectors, especially for their potential application in functional foods and as probiotics, is still necessary. Moreover, considering the remarkable intra-individual and inter-individual differences in the conversion rate of lignans into enterolignans, further efforts must be made for the development of a more accurate protocol for qualitative and quantitative analysis of these metabolites in human biofluids. For instance, to our knowledge, no validated protocols are currently available for the detection/quantification of enterolignans in human fecal samples. These future methodological developments could allow for the innovative enterolignans use as novel reliable and non-invasive biomarkers of GM dysbiosis and associated (intestinal and systemic) human diseases.

## Figures and Tables

**Figure 1 molecules-28-00343-f001:**
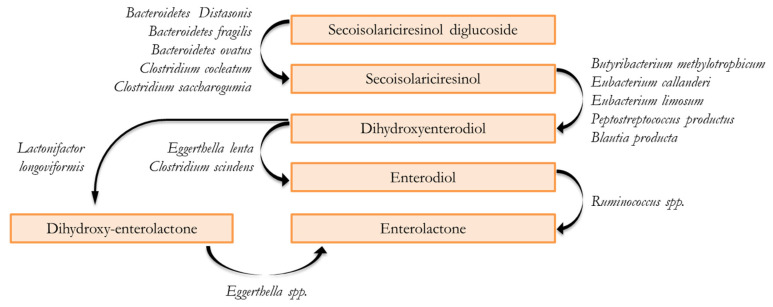
Schematic representation of the fermentative process of lignans.

**Figure 2 molecules-28-00343-f002:**
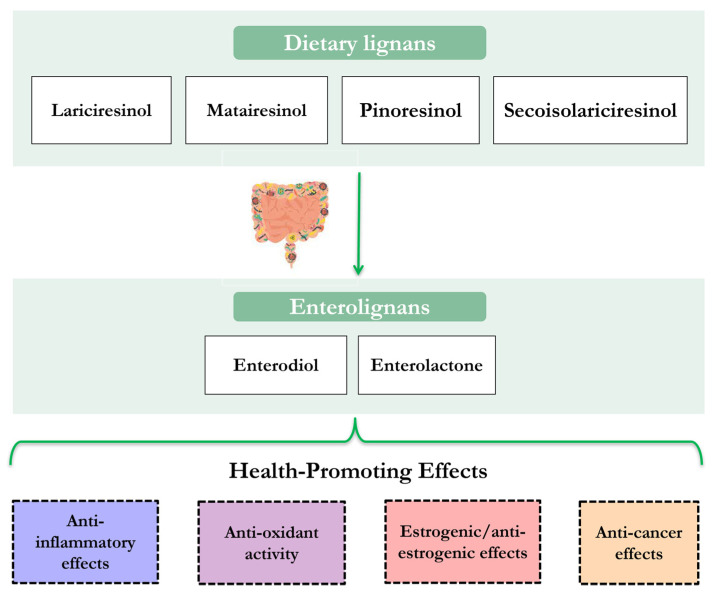
Main health-promoting proprieties of microbial-derived enterolignans.

**Figure 3 molecules-28-00343-f003:**
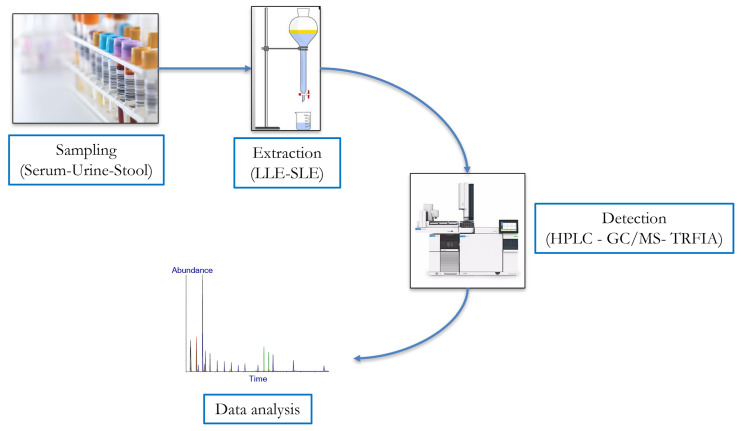
Experimental workflow for qualitative and quantitative analysis of lignans. LLE: liquid-liquid extraction, SLE: solid-liquid extraction, HPLC: high-performance liquid chromatography, GC/MS: gas chromatography mass spectrometry, TRFIA: time-resolved fluoro-immunoassay.

**Table 1 molecules-28-00343-t001:** Main advantages, disadvantages and applicability of the leading lignans detection methods. HPLC: high-performance liquid chromatography, GC/MS: gas chromatography mass spectrometry, TRFIA: time-resolved fluoro-immunoassay

Detection Method	Advantages	Disadvantages	Applicability
HPLC	Reduced sample pre-treatment High selectivity, resolution, speed, sensitivity and reproducibility of analyses	No universal detector Less separation efficiency than GC/MS More difficult for novices	Human and animal biofluids Plant matrixes
GC/MS	Allows the identification of multicomponent mixtures Provide unambiguous qualitative and quantitative information	Requires high temperatures, that can damage the analytes Difficult sample pre-treatment (e.g., sample derivatization)	Human and animal biofluids Plant matrixes
TR-FIA	High speed and sensitivity	Very low selectivity since cross-reactivity	Human biofluids

## Data Availability

Data sharing not applicable. No new data were created or analyzed in this study. Data sharing is not applicable to this article.

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
