# Peer review of "Interplay between Lignans and Gut Microbiota: Nutritional, Functional and Methodological Aspects"

_molecules, 2023, doi:10.3390/molecules28010343_

Round 1

Reviewer 1 Report

The authors provide an overview related to enterolignans formed by action of gut microbes on dietary lignans, the biological properties and the methodologies to quantitate enterolignans were also discussed. The paper is well written but it will be even better if authors incorporate the mechanism of enterolignans for prevention of cancer and cardiovascular diseases. Please find below the suggestions or queries that are needed to be considered:

1. Line106, write the full form of END.

2. Fig. 1 diagram, Healt should be replaced by health.

3. Line205, hight should be replaced by high.

4. Line210, citation (Ren 2017) should be numbered as per the citation style.

Author Response

REVIEWER 1

 Q/S 1: The paper is well written but it will be even better if authors incorporate the mechanism of enterolignans for prevention of cancer and cardiovascular diseases.

 Q/S 1 Reply: As rightly suggested by the reviewer we have introduced the mechanism of enterolignans for prevention of cancer (please see lines 211-219 and 233-237) and cardiovascular diseases (please see lines 243-246)

Q/S 2: Line106, write the full form of END.

Q/S 2 Reply: Thank you, we are sorry for the mistake, we mean ED (enterodiol) and not END and we have correct it in the text.

Q/S 3: Fig. 1 diagram, Healt should be replaced by health.

Q/S 3 Reply: Thank you, we are sorry for the mistake, we have corrected it in Figure 1.

Q/S 4: Line205, hight should be replaced by high.

Q/S 4 Reply: Thank you, we are sorry for the mistake, we have corrected it in the text.

Q/S 5: Line210, citation (Ren 2017) should be numbered as per the citation style.

Q/S 5 Reply: Thank you, we have corrected the mistake in the text, the citation of Ren and colleagues is the reference number 85.

Reviewer 2 Report

This review aims to explore the beneficial roles of enterolignans, their interaction with gut microbiota and the new methodological approaches developed for their evaluation in different matrix. The title also reflects these aims, however the content of the review does not represent these topics.

Some pieces of advises to improve the document:

Abstract:

-The authors claim that lignans are polyphenols present in a wide range of foods consumed daily. This affirmation, it is not true. In fact, lignans are a very particular family of polyphenols which are found in specific sources (seeds, beans and some soy products), and they are not daily consumed at all.

-What do you mean by lignans modulate enzyme activity?

Manuscript:

In general terms, the information of the introduction is really poor and not well connected with the topics of the review. Moreover, next comments should be considered.

-       The sentence “potent antioxidant and anti-carcinogenic activities and ii) estrogen agonism/antagonism properties. In addition, recent evidence have highlighted their beneficial role in the prevention and treatment of menopausal symptoms, skin aging, osteoporosis, cancer, cardiovascular, neurodegenerative, immune and metabolic diseases” is not according to reference 3 as cited in the manuscript.

-       When you describe that some interindividual variations could be observed in enterolignan production, you should described the importance of interindividual differences in gut microbiota since this is the most important factor. Please check the review ( https://doi.org/10.1039/d1fo02033a)

-What is the argument to claim that the health benefits from lignans are mainly from lariciresinol, matairesinol, pinoresinol, and secoisolariciresinol (SECO)?

-When you say that flax and sesame seeds provide 284.00mg/100g and 776.49 mg/100g respectively. All varieties are the same? These are general values?

- In the section of “gastrointestinal digestion and fermentative process of lignans”, the metabolism of the lignans is poor described and some relevant reference should be given (https://doi.org/10.1111/j.1574-6941.2007.00330.x)

- In the figure1 the main health-promoting proprieties of microbial-derived enterolignans are schematized, however all of these activities are not discussed in the main text (e.g., estrogenic and antiestrogenic capabilities).

-Regarding the techniques to analyze lignans and enterolignans they should be sum up and give a more critical point of view comparing the different approaches used.

Author Response

Abstract

Q/S 1: The authors claim that lignans are polyphenols present in a wide range of foods consumed daily. This affirmation, it is not true. In fact, lignans are a very particular family of polyphenols which are found in specific sources (seeds, beans and some soy products), and they are not daily consumed at all.

Q/S 1 Reply: In agreement with the reviewer suggestion, considering that not in all countries lignans are daily consumed, we have modified the sentence (line 14).

Q/S 2: -What do you mean by lignans modulate enzyme activity?

Q/S 2 Reply: We thank the reviewer for the question. For instance, lignans have been described as strong antioxidants through the enhancement of antioxidant enzymes activity (DOI: https://doi.org/10.1016/j.tifs.2020.10.015) or, by altering the activity of enzymes involved in estrogen metabolism, lignans can change the biological activity of endogenous estrogens (DOI: 10.1016/j.jsbmb.2005.02.002)

Manuscript

Q/S 3: The sentence “potent antioxidant and anti-carcinogenic activities and ii) estrogen agonism/antagonism properties. In addition, recent evidence have highlighted their beneficial role in the prevention and treatment of menopausal symptoms, skin aging, osteoporosis, cancer, cardiovascular, neurodegenerative, immune and metabolic diseases” is not according to reference 3 as cited in the manuscript.

 Q/S 3 Reply: In agreement with the reviewer we have change the reference 3 with PMID: 31130604

Q/S 4: When you describe that some interindividual variations could be observed in enterolignan production, you should described the importance of interindividual differences in gut microbiota since this is the most important factor. Please check the review ( https://doi.org/10.1039/d1fo02033a)

Q/S 4 Reply: In agreement with the reviewer, we have introduced the pivotal role of GM as a main   driver of the interindividual variation (please see lines 133-137)

Q/S 5: What is the argument to claim that the health benefits from lignans are mainly from lariciresinol, matairesinol, pinoresinol, and secoisolariciresinol (SECO)?

Q/S 5 Reply: We thank the reviewer for the appropriate question. Lariciresinol, matairesinol, pinoresinol and secoisolariciresinol are the lignans most present in foods and for which there is the largest number of studies demonstrating their potential health benefits. Although current extraction techniques have allowed the identification of a large number of other lignans, information on their beneficial effects fot host’ health are still very limited and more studies are needed.

Q/S 6: When you say that flax and sesame seeds provide 284.00mg/100g and 776.49 mg/100g respectively. All varieties are the same? These are general values?

Q/S 6 Reply: We thank the reviewer for the right question. All values reported in the paper come from the latest version of the Phenol-Explorer, a comprehensive database on polyphenol content in foods developed by the French National Research Institute for Agriculture, Food and the Environment (INRAE) with support from scientific communities such as the University of Alberta, the University of Barcelona and the International Agency for Research on Cancer (IARC). The data contained in the database is derived from the systematic collection of more than 60,000 original content values reported in more than 1,300 scientific publications. In detail, the values reported in our manuscript are the mean content of lignans in foods.

Q/S 7: In the section of “gastrointestinal digestion and fermentative process of lignans”, the metabolism of the lignans is poor described and some relevant reference should be given (https://doi.org/10.1111/j.1574-6941.2007.00330.x)

Q/S 7 Reply: In agreement with the reviewer, we have introduced the relevant reference https://doi.org/10.1111/j.1574-6941.2007.00330.x (please see lines 85-91)

Q/S 8:  In the figure1 the main health-promoting proprieties of microbial-derived enterolignans are schematized, however all of these activities are not discussed in the main text (e.g., estrogenic and antiestrogenic capabilities).

Q/S 8 Reply: As rightly suggested by the reviewer we have discussed the anti-inflammatory,anti-oxidant (please see lines 178-187) anti-cancer (please see lines 211-219 and 233-237) and estrogenic and anti-estrogenic (please see lines 144-158) activities of enterolignans

Q/S 9:  Regarding the techniques to analyze lignans and enterolignans they should be sum up and give a more critical point of view comparing the different approaches used.

Q/S 9 Reply: As rightly suggested by the reviewer, we have prepared a Figure recapitulating of the experimental workflow for qualitative and quantitative analysis of lignans (please see Figure 3) and summarised the uses of the different approaches (please see lines 392-398).

Reviewer 3 Report

In this manuscript, authors reviewed the beneficial roles of lignans, their interaction with gut microbiota and methodological approaches for their evaluation. The review is comprehensive and specific, but some issues need to be addressed before publication.

1. The manuscript summarized gastrointestinal digestion and fermentative process of lignans. Authors described the interactions between dietary lignans and gut microbiota, and it’s suggested to be shown in a Figure, which can be more concise and clearer.

2. Lignans have many biological functions and can reduce the risk on various pathologies, such as cancers, cardiovascular diseases. But the specific molecular mechanisms are not well understood. Is the current research too few to be summarized, or not complete review, which needs to be further in-depth.

3. It is suggested that the article use charts to make the content more vivid and easy to understand.

Author Response

Q/S 1: The manuscript summarized gastrointestinal digestion and fermentative process of lignans. Authors described the interactions between dietary lignans and gut microbiota, and it’s suggested to be shown in a Figure, which can be more concise and clearer.

Q/S 1 Reply: As rightly suggested by the reviewer, we have prepared a Figure summarising of the fermentative process of lignans (please see Figure 1).

Q/S 2: Lignans have many biological functions and can reduce the risk on various pathologies, such as cancers, cardiovascular diseases. But the specific molecular mechanisms are not well understood. Is the current research too few to be summarized, or not complete review, which needs to be further in-depth.

Q/S 2 Reply: As rightly requested by the reviewer, although few information are available in the current literature about the molecular mechanism of lignans in the reduction on the risk of different pathologies , we have introduced the mechanism of enterolignans for prevention of cancer (please see lines 196-204 and 218-222) and cardiovascular diseases (please see lines224-231)

Q/S 3: It is suggested that the article use charts to make the content more vivid and easy to understand.

Q/S 3 Reply: As rightly suggested by the reviewer, we have also prepared a Figure summarising of the experimental workflow for qualitative and quantitative analysis of lignans (please see Figure 3).

Round 2

Reviewer 3 Report

Overall, this is an improved manuscript, where the authors largely address Reviewers’ concerns. I just have a few quick questions.

1. Line 222-231: the reference should be supplemented.

2. Line 328: “thanks to the EL capability” was no need to italics.

3. Part 5: the advantages, disadvantages and applicability of various detection methods can be compared in the table.

Author Response

Q/S 1: Line 222-231: the reference should be supplemented

Q/S1 Reply: We thank the reviewer for the right suggestion, we have introduced in the text the references number 52 and 53

Q/S 2: Line 328: “thanks to the EL capability” was no need to italics.

Q/S2 Reply: Thank you, we have corrected the mistake in the text (please see line 233)

Q/S 3: Part 5: the advantages, disadvantages and applicability of various detection methods can be compared in the table.

Q/S3 Reply: In according to reviewer suggestion we have prepared a Table (Table 1) summarizing the main  advantages, disadvantages and applicability of  the different detection methods